# Label-Free, High-Throughput Assay of Human Dendritic Cells from Whole-Blood Samples with Microfluidic Inertial Separation Suitable for Resource-Limited Manufacturing

**DOI:** 10.3390/mi11050514

**Published:** 2020-05-19

**Authors:** Mohamed Yousuff Caffiyar, Kue Peng Lim, Ismail Hussain Kamal Basha, Nor Hisham Hamid, Sok Ching Cheong, Eric Tatt Wei Ho

**Affiliations:** 1Department of Electrical & Electronics Engineering, Universiti Teknologi PETRONAS, Perak 32610, Malaysia; yousuff.ece@cahcet.edu.in (M.Y.C.); ismailhussain22@gmail.com (I.H.K.B.); hishmid@utp.edu.my (N.H.H.); 2Department of Electronics and Communication Engineering, C. Abdul Hakeem College of Engineering and Technology, Melvisharam, Tamil Nadu 632509, India; 3Head and Neck Cancer Research Group, Cancer Research Malaysia, Selangor 47500, Malaysia; kuepeng@gmail.com (K.P.L.); sokching.cheong@cancerresearch.my (S.C.C.); 4Department of Oral & Maxillofacial Clinical Sciences, Faculty of Dentistry, University of Malaya, Kuala Lumpur 50603, Malaysia

**Keywords:** inertial spiral microfluidics, dendritic cell separation, resource-limited microfluidics

## Abstract

Microfluidics technology has not impacted the delivery and accessibility of point-of-care health services, like diagnosing infectious disease, monitoring health or delivering interventions. Most microfluidics prototypes in academic research are not easy to scale-up with industrial-scale fabrication techniques and cannot be operated without complex manipulations of supporting equipment and additives, such as labels or reagents. We propose a label- and reagent-free inertial spiral microfluidic device to separate red blood, white blood and dendritic cells from blood fluid, for applications in health monitoring and immunotherapy. We demonstrate that using larger channel widths, in the range of 200 to 600 µm, allows separation of cells into multiple focused streams, according to different size ranges, and we utilize a novel technique to collect the closely separated focused cell streams, without constricting the channel. Our contribution is a method to adapt spiral inertial microfluidic designs to separate more than two cell types in the same device, which is robust against clogging, simple to operate and suitable for fabrication and deployment in resource-limited populations. When tested on actual human blood cells, 77% of dendritic cells were separated and 80% of cells remained viable after our assay.

## 1. Introduction

Throughout the 30-year development history of microfluidics technologies [1], a broad array of inventive concepts, novel physics and prototype devices have been demonstrated which miniaturize fluidic manipulation to achieve large-scale simultaneous assays with nanoliter reactions and precise manipulation and inspection of microscopic objects [2]. While microfluidics technology has emerged as a powerful laboratory tool for academic research, there is an absence of its mass adoption toward solving compelling commercial or communal problems. In healthcare, microfluidics technology has a pressing opportunity to enable point-of-care (POC) testing of infectious diseases and improve healthcare services and patient outcomes in places with limited laboratory infrastructure. Microfluidics POC devices may enable novel ways to monitor population health and improve accessibility to medical interventions delivered into bodily fluids. Cell assays which count the number of circulating white blood cells could identify when a person is fighting an infection or inflammation, possibly before symptoms manifest. Microfluidic lab-on-chips could stimulate immune cells for immunotherapy [3], which is an emerging treatment for cancer tumors.

For microfluidic devices to be widely adopted in resource-limited communities, the World Health Organization (WHO) recommends that devices fulfill the ASSURED criteria, to be Affordable, Sensitive, Specific, User-friendly, Rapid and Robust, Equipment-free and Delivered to the people who need it [4]. Microfluidics technology was developed to miniaturize lab-scale processes and reactions into small portable devices for rapid processing of small fluid volumes amenable for POC operation [5]. Most inventions were focused on advancing miniaturization or improving precision control over physical forces [6]. However, innovations to improve affordability, robustness and ease of use have been neglected, and microfluidic devices depend on elaborate laboratory setups. Prototypes are incompatible with industrial-scale fabrication [1], and they are not easy to operate by untrained personnel; therefore, they are not accessible to resource-limited populations. Paper microfluidics [7,8,9] is the only major innovation, in recent years, to address the need for affordable and simple microfluidic devices to deliver POC health services in the developing world.

In this paper, we develop an alternative approach, using inertial microfluidics to separate immune cells from blood and to select dendritic cells. There are several distinct types of white blood cells which trigger or mount innate or adaptive immune responses in the body. Cell assays that uniquely quantify the distinct white blood cell populations can yield insights into the state of an individual’s immune system and health. Dendritic cells are rare immune cells that react toward foreign antigens, to trigger the adaptive immune response [10]. We are especially interested in isolating dendritic cells because these cells may be stimulated with tumor-associated antigens to initiate CD8+ T lymphocytes (CD8-T) assault on tumors, as an immunotherapy called dendritic cell vaccination (DCV) [11,12]. For interventions like DCV, where cells must be returned to the human body, microfluidic cell assays must operate on reagent- and label-free principles. Excluding the use of reagents and labels also significantly improves the affordability and ease of operating devices at the POC, since it obviates the logistics of purchasing, transporting, qualifying and measuring accurate volumes of chemicals, antibodies or beads.

Adequate quantities of dendritic cells for clinical studies are commonly obtained by differentiating precursor monocytes in Peripheral Blood Mononuclear Cells (PBMC) extracted by leukapheresis [13] in culture with cytokines [14]. Despite numerous careful studies showing similarity in phenotype and function with natural dendritic cells, doubts prevail over the efficacy of monocyte-derived dendritic cells which may be compromised by the culture process or reagents [12]. Natural dendritic cells enriched from leukapheresis product successfully induced antitumor immune response in clinical studies of melanoma and prostate cancer [15,16]. Isolating dendritic cells circulating in the bloodstream for immunotherapy appears to be efficacious, even at doses at an order of magnitude less cells. Dendritic cells can be enriched from PBMC, using immuno-magnetic isolation with a yield between 20% and 80% purity [17]. 

If dendritic cells can be enriched without magnetic labeling or reagent use [17], lab-based separation equipment would no longer be required, and the entire cold chain to ensure that whole blood extracted from the patient is free from degradation or contamination during transport to the lab could be eliminated. High-throughput, label- and reagent-free enrichment would remove infrastructure and expertise barriers and make dendritic cells accessible at the point-of-care (POC), and perhaps even enable dendritic cell antigen loading at the POC.

Microfluidic devices utilizing inertial forces are capable of high-throughput, label- and reagent-free cell selection [18,19,20,21,22]. Inertial microfluidics utilize the shape of fluidic channels, to induce forces arising from fluidic flow, to separate cells. Inertial devices are easy to use and require almost no additional supporting equipment. Aside from the microfluidic device, only a small portable syringe pump is required to inject cell-containing fluids, at a consistent flow rate, into the device. Canonical spiral inertial microfluidic designs use narrow channels (diameter < 150 µm) to separate cells into distinct focused streams [19,22,23,24,25,26]. Narrower channels generate larger forces to part cells into streams with larger separation distances. However narrow channels are not robust; inertial forces cannot separate all blood cells if there is a broad variety of cell sizes and narrow channels are prone to clogging. Furthermore, a slight manufacturing imprecision in the channel width relative to the intended design dimension will affect device performance (for example, a manufacturing variability of ±10 µm in a 100 µm width channel leads to significant deviation from intended performance). For this reason, inertial devices have been manufactured with high-precision lithographic manufacturing [27] and require stringent quality control. This fabrication method requires significant investment in infrastructure and skilled operators which can only be amortized by selling large quantities of the same device. It is prohibitive for resource-limited populations because the barriers to entry are too high.

We present our innovations to spiral inertial microfluidic device design that effectively separates blood cells sized between 6 and 15 µm into three separate streams. Our novel design utilizes larger channel sizes (200 to 600 µm) and is fabricated with the low-cost xurography [28,29] method, whereby a mask is cut from a polymer sheet. Using fluorescence beads, we validate that our device selects particles by size and subsequently validate the selection of white blood cells and dendritic cells, as well as cell viability, using flow cytometric analysis. 

## 2. Materials and Methods 

### 2.1. Device Design and Simulation Studies

Our device employs a spiral microfluidic channel with 7 turns, 100 µm height and gradually increasing width from 200 µm at the inlet to 600 µm at the outlet, with spiral radius varying from R = 2 mm to R = 6.2 mm (Figure 1a(i)). The three-dimensional spiral geometry of the device design was simulated with computerized fluid dynamics (CFD) software (Ansys Fluent Inc., Canonsburg, PA, USA). We solved the Navier–Stokes equations for flow fields without particles, using a Third-Order MUSCL scheme and the SIMPLEC algorithm. Using this computational fluid dynamic simulation [30], we verified that our design would route particles of 15, 10 and 7 µm mean diameter into Outlets 1, 2 and 3, respectively, because we found that human dendritic cells are 10–15 µm, white blood cells are 7–12 µm and red blood cells are 6–8 µm in size (Figure 2). Outlet 4 collects excess blood plasma. 

### 2.2. Device Fabrication in Resource-Limited Settings

The spiral microchannel was fabricated by using a low-cost technique, xurography. The desired pattern was designed with a CAD software (AutoCAD version X, Autodesk Inc., SAN Rafael, CA, USA), and the pattern was cut into an adhesive vinyl sheet. The desired microchannel was removed from the sheet and filled with epoxy polymer. The epoxy polymer was cured for 20 h, and the inverse pattern of the microchannel was obtained in epoxy, which was then used as a mold. The epoxy mold was used to manufacture polydimethylsiloxane (PDMS) spiral devices by replica molding. A mixture of PDMS base (Sylgard 184 base, Dow Corning, Midland, MI, USA) and curing agent (10:1 ratio) (Sylgard 184 curing agent, Dow Corning, Midland, MI, USA) was cast on epoxy master and cured for 24 h, at room temperature. Cured PDMS devices embedded with channels were peeled off from the master mold, and inlet/outlet ports were punched, using a 2 mm biopsy punch. The PDMS device was cleaned with cellulose tape (Scotch Tape, 3M, St. Paul, MN, USA) and ethanol. Subsequently, the device was permanently bonded to a standard glass slide, using UV ozone system (Nova Scan Inc., Ames, IA, USA), for 5 min. The UV-treated device was immediately baked in a hot plate, at 80 °C, for 15 min, to increase the bonding strength.

### 2.3. Device Validation with Assay of Polystyrene Microparticles

Suspended 15 µm diameter green fluorescent, 10 µm non-fluorescent and 7 µm diameter red fluorescent polystyrene microspheres (Bangs Laboratories) were mixed and diluted in deionized water, with ~500,000 particles/mL. The mixture was then introduced into spiral microchannel, using a programmable syringe pump (New Era NE-1000, New Era Pump Systems, Inc., Farmingdale, NY, USA) connected to the microchannel inlet with silicone tubing. Focused particle streams were visualized with an inverted microscope (Nikon Ti-S Microscope, CFI Plan Fluor 4X objective, Nikon Singapore Pte Ltd., Singapore). Fluorescence and brightfield microscope images were captured with a 12-bit CCD Camera (Retiga Exi, QImaging, Surrey, BC, Canada).

### 2.4. Blood Cells Sample Preparation

Blood samples were collected after informed consent and with approval by the Ethics Committee, Faculty of Dentistry, University of Malaya, DF OS0910/0049 (L). Freshly isolated Peripheral Blood Mononuclear Cells (PBMC) from the blood samples were cultured in M-SFM media and incubated for 1 h, at 37 °C. Then the plates were shaken, to loosen all non-attached cells, and these were aspirated and removed. The adherent cells remaining on the plate were then cultured with M-SFM, in the presence of 100 ng/mL GM-CSF and 25 ng/mL IL-4, to induce differentiation of dendritic cells. The cells were incubated in culture media for 5 days, to grow and differentiate into mature dendritic cells. Mature DCs were trypsinized and centrifuged, to remove the supernatant, and final DC count obtained was at ~35,000 cells/mL. Then, 2 mL of DCs was mixed with 3 mL of diluted white and red blood cells (1000× dilution). 

### 2.5. Blood Cell Selection Assay

The mixture of blood cells was injected into the device, with a syringe pump, at a constant flow rate of 1.9 mL/min and collected into 4 different outlets. Samples from each collection reservoir were stained with BD Horizon™ Fixable Viability Stain 780 (FVS780) viability dye (BD Biosciences Cat No. 565388), for 15 min, at room temperature, followed by adding an antibodies mixture comprised of Lineage cocktail 1 (BD Biosciences Cat No. 340546), PE conjugated mouse anti-human CD123 (BD Biosciences, Cat No 340545) and APC conjugated Mouse anti-human CD11c (BD Biosciences Cat No 340544) and further incubated for 30 min, at 4 °C. Following incubation, the cells were washed with phosphate buffer saline prior to flow cytometer analysis (BD FACSCanto II, BD Biosciences, Piscataway, NJ, USA). The flow cytometer was used to quantify the purity and recovery rate of dendritic cells from our device. Viable dendritic cells are the sum of cells with CD11c+ and CD123- expression (typically myeloid DCs), cells with CD123^+^ and CD11c^-^ expression (typically plasmacytoid DCs) and cells expressing both markers.
(1)$$\mathrm{Recovery}\;\mathrm{rate}=\frac{\mathrm{number}\;\mathrm{of}\;\mathrm{DCs}\;\mathrm{in}\;\mathrm{one}\;\mathrm{outlet}}{\mathrm{number}\;\mathrm{of}\;\mathrm{DCs}\;\mathrm{in}\;\mathrm{all}\;\mathrm{outlets}}$$

With the aid of a flow cytometer, we used FVS780 for discrimination of viable from non-viable mammalian cells. Blood cell mixture before and after processing by our spiral microfluidic device was mixed with FVS780 and quantified, using BD FACS Canto II (BD Biosciences, USA).
(2)$$\mathrm{Viability}\;\mathrm{rate}=\frac{\mathrm{number}\;\mathrm{of}\;\mathrm{viable}\;\mathrm{blood}\;\mathrm{cells}\;\mathrm{in}\;\mathrm{each}\;\mathrm{outlet}\;}{\mathrm{total}\;\mathrm{cell}\;\mathrm{events}\;\mathrm{in}\;\mathrm{each}\;\mathrm{outlet}\;}$$

### 2.6. Statistical Analysis

Statistical analysis was conducted, using GraphPad Prism 5. One-way ANOVA was used to compare the differences of viability and DC recovery from Outlets 1–4. A *p* < 0.05 was considered to be significant. 

## 3. Results

### 3.1. A Spiral Microfluidic Device with Channels Widths Exceeding 200 μm, for Focused Fractionation of Multiple Particle Sizes

We designed a spiral microfluidic device to separate 15, 10 and 7 µm beads. Our device has channel widths exceeding 200 μm, making fabrication possible with a low-cost xurography method. A prototype of the device is shown in Figure 1a, which includes Scanning Electron Microscope (SEM) images of the planar and cross-sectional profiles. PDMS molds produced with xurography produce smooth channel walls (Figure 1a, sections ii and iii) with a manufacturing tolerance of less than 5% (Table 1). The 15, 10 and 7 µm particle streams are successfully collected into Outlets 1, 2 and 3, respectively, despite being closely spaced by approximately 20 µm (Figure 1b). Measurements at the collection outlet reveal that 99% of the 15 µm beads, with 8% of 10 µm and 1% of 7 µm beads, were routed into Outlet 1. The remaining 92% of 10 µm beads were routed into Outlet 2, together with 8% of the 7 µm beads [30].

To collect the closely spaced particle streams, we designed the fluidic resistance of the collection outlets such that particles greater than 13 µm would flow into Outlet 1; particles between 8 and 11 µm would flow into Outlet 2; and particles between 6 and 7 µm would flow into Outlet 3. Fluidic resistance was tuned with computational fluid dynamic simulations (Figure 2a), by changing the collection channel length without changing channel width. To ensure inertial focusing of all particles, the spiral channel width was gradually enlarged from 200 to 600 μm, to hasten the decay of Dean drag forces from inlet to outlet (Figure 2b). Under a constant flow rate, the expanding channel width changes the fluid velocity and particle Reynolds number at different loci of the spiral (Table 2).

### 3.2. Red, White Blood and Dendritic Cell Separation Assay

Our spiral microfluidic device was designed to separate red blood cells (RBC), white blood cells (WBC) and dendritic cells (DC). We performed experimental measurements from images of blood cells, to obtain the distribution of red, white blood and dendritic cell sizes (Figure 3a). We found that the majority of red blood cells are between 6 and 7 μm; white blood cells are between 8 and 11 μm; and dendritic cells are between 13 and 15 μm. We also found that our device does not significantly damage cells, despite the high flow rates required for inertial microfluidics. More than 80% of cells passing through all four collection outlets remained viable (Figure 3c). There is no significant difference in cell viability for cells passing through Outlet 1, Outlet 2, Outlet 3 and Outlet 4 (81.8%, 91.5%, 91.9% and 89.4% respectively).

When tested with fluid containing a single blood-cell type alone, at 100× dilution, we observed that the majority of dendritic cells are collected in Outlet 1 and majority of red blood cells are collected in Outlet 3 (Figure 4a). Although more white blood cells are collected in Outlet 2, a significant fraction is also present in Outlet 3. This reflects the distribution of white blood cell sizes, which overlaps with red blood cells (Figure 3b) rather than the failure of the inertial spiral microfluidic device itself. We subsequently evaluated the separation efficiency of dendritic cells alone (Figure 4b), tested with a fluid containing all three blood cell types. The collection efficiency of dendritic cells is lower, at 77.2%, in Outlet 1.

### 3.3. Device Performance under Variations in Flow Rate and Cell Concentration 

Inertial microfluidic devices must be operated above a threshold flow rate, for inertial forces to become effective. We performed simulations of our spiral device at different fluid velocities, to determine the range of flow rates that guarantees particle focusing and separation. Our simulations show that our device will perform as desired when operated at flow rates between 1.5 and 1.9 mL/min. At flow rates below 1.5 mL/min, Dean drag forces are not sufficiently strong to sustain size-dependent lateral migration of particles. At flow rates above 2.0 mL/min, Dean drag forces dominate inertial lift forces, and turbulent lateral flow disrupts inertial focusing (Figure 5).

We tested the cell-separation efficiency of our device on WBC only, using different flow rates (Figure 6a). Our experimental results concur with the simulations, and at 1.6 mL/min, the largest proportion of WBC (88%) was collected in Outlet 2. We also tested the effect of different cell concentrations on the inertial separation behavior with RBC only. Experimental results show that dilutions of 100× to 200× yield optimal collection efficiency at Outlet 3 (Figure 6b).

## 4. Discussion

Spiral inertial devices induce inertial forces within a fluidic flow, due to channel geometry, curvature ($R_{c}$) and high fluidic flow rates [31,32,33]. Two main forces act on cells (with size $a_{p}$) flowing with the fluid, lift force $\left(F_{L}\propto \frac{a_{p}{}^{4}}{wh}\right)\left(w:\;\mathrm{width}\;\mathrm{and}\;h:\;\mathrm{height}\right)\;$[34] and dean force $\left(F_{D}\propto \frac{a_{p}}{R_{c}}\right)$ [31], such that the balance between these forces migrates the cells to stable flow positions, in focused and distinct streamlines. The ideal condition for cell separation is when the dean drag is on the same order as the inertial lift $\left(F_{L}\geq F_{D}\right)$, such that cells of different sizes experience different equilibrium forces and occupy size-dependent displacement across the channel cross-section. The lift force focuses cells, while the dean force separates cells into distinct streams. Two limiting cases should be avoided: (1) *F_L_* ≫ *F_D_*, cell migration is dominated solely by inertial lift and cells of all sizes are focused to the same equilibrium position; (2) *F_L_* ≪ *F_D_*, the dominant dean force mixes cells through cross-sectional channel flow and no focusing occurs. 

Archetypical implementations of spiral separation devices use small channel widths relative to cell size, to generate sufficient inertial lift force for focusing, because lift force is inversely proportional to channel dimensions $\left(F_{L}\propto \frac{a_{p}{}^{4}}{wh}\right)$. The magnitude of the inertial forces may be designed to be significantly larger than necessary, to create a sufficient gap between focused cells streams, since the equilibrium position of cells of different sizes is dependent on the ratio of the inertial lift to dean force, FLFD=(aph)3 [18,32] require precision fabrication techniques and economies of scale for cost efficiency. 

However, narrow channel widths impose problems downstream, when attempting to collect separated cells streams. When cells from a broad range of sizes must be separated, it becomes challenging to maintain the lift-force-to-dean-force ratio for cells of all sizes, if small channel widths are used, as some cells will inevitably become defocused and mix into other focused streams. There are insufficient degrees of freedom in the geometrical design parameters to enhance separation distance between multiple cell streams if small channel widths are used. Narrow-width spiral devices cannot be fabricated without high-precision lithographic techniques [35], which require significant infrastructure that cannot be easily set up in resource-limited settings. In the small-channel width regime, only small variations in channel width dimensions can be tolerated, or else the fabricated device will rapidly lose efficiency and not perform as intended [23,36].

Conventionally, the channel at the device outlet is divided into multiple outlets of narrower widths, to collect the focused cell streamlines into different outlets [18,23] If many cell streams must be collected, or if there is a small size difference between cells streams, the separation distance between two distinct cell streamlines can be very closely spaced (<50 µm) [18] due to channel width constraints and become difficult to collect separately. Larger cells may clog the narrow outlets or become damaged from the impact against the outlet walls. Expanding outlets have been designed to ameliorate the problem, but this causes the cells to defocus and mix. 

Our design overcomes all these challenges, and we demonstrate separation of multiple cell types (red blood cells, white blood cells and dendritic cells), without reagents or labels, for true point-of-care operation. Our main innovation is in making spiral inertial designs work with larger-than-conventional channel widths. We used a novel collection channel design which can route closely spaced cell streamlines (~20 µm) into different collection channels, with high accuracy [30]. Without having to generate large forces to increase particle stream separation, we gain an additional degree-of-freedom in the design, which enables us to use larger channel widths.

Although this runs against the prevailing trend in the microfluidics community toward miniaturization, using channel widths greater than 200 µm confers several distinct advantages. To our knowledge, ours is the first demonstration of how to sustain more than two separated focused streams of cells in spiral inertial devices. With larger channel widths, we are able to balance the inertial forces so that inertial focusing and separation effects apply to cells between 7 and 15 µm in size. Microfluidic designs using channels exceeding 200 µm in width are more tolerant of manufacturing variations and can also be fabricated with less-precise manufacturing methods, like xurography. Designs with larger channel widths are less susceptible to clogging or damaging cells.

We designed the spiral device such that inertial forces dominate Dean forces for the largest particles. Therefore, the largest particles will undergo inertial focusing first and gradually migrate laterally to their equilibrium position, moving through the spiral. Inertial lift forces decay, and Dean forces become more dominant. With this configuration, however, the smallest particles will experience dominant Dean forces at the inlet. In conventional fixed-width spirals, the Dean forces’ magnitude does not change much throughout the spiral. By introducing an expanding channel width to the spiral, we introduce a controllable decay in Dean force, by gradually reducing the particle Reynold’s number. This allows inertial lift forces to dominate and promotes focusing of the smallest particles at spiral turns closer to the outlet.

The balance of inertial lift and dean forces in spiral devices are sensitive to changes in flow rate. Lift force is proportional to the square of flow rate, whereas Dean force is inversely proportional to flow rate (through the particle Reynolds number). Simulation studies show that our design functions well between flow rates of 1.5 and 1.9 mL/min. Reducing the flow rate beyond that range results in a rapid loss of inertial lift force and a corresponding rapid increase in Dean force. Increasing the flow rate beyond that range is also not favorable. Besides displacing the equilibrium positions of the particle streamlines, lateral flow may become turbulent as Dean velocity increases and disrupt particle focusing. 

Unlike fluorescent beads, blood cells are deformable. Cell deformation may induce an additional lift force which acts in the direction of the channel center [37]. Differences in cellular deformability have been used to separate and enrich malaria-infected red blood cells from healthy red blood cells and circulating tumor cells from peripheral blood. Several studies have demonstrated that the rigid polystyrene beads can appropriately mimic and predict the inertial focusing behavior of the blood cells [38,39,40] 

Our device is portable and simple to operate, as it only requires a small syringe pump and can be fabricated and deployed in low resource settings. Only careful design of the channel geometry and operation at high fluid flow rates, ~1.9 mL/min, suffice. High throughput operation is an automatic feature. This is because high flow rates are necessary to induce inertial forces for proper device operation. White blood cells and dendritic cells remained viable after enrichment and size-based separation successfully isolates dendritic cells from blood and a significant portion of the white blood cells. 

Our experiments with fluorescent beads confirm the size-selective function of our microfluidic device. However, large counts of both red and white mononuclear blood cells are also routed into Outlet 1, and cells were also routed to Outlet 4, which is not an intended feature of our design. A likely cause is the deterioration of inertial focusing due to dense cell concentrations. At cell-to-volume fractions between 1% and 3%, inertial forces are known to align cells laterally into a single stream and to regulate interstream spacing [32,41]. Above that fraction, inertial focusing deteriorates due to interparticle interactions [42]. We anticipate significant improvements in performance if we sufficiently dilute the blood samples [34,43].

## Figures and Tables

**Figure 1 micromachines-11-00514-f001:**
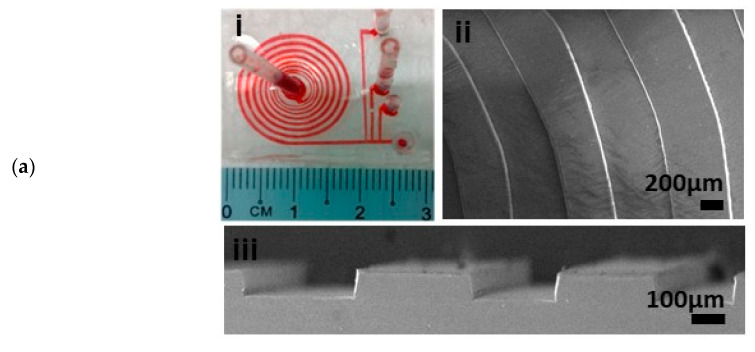
An inertial spiral microfluidic device for separating fluorescent beads of 3 different sizes: (**a**) (**i**) a PDMS device is designed with channel widths exceeding 200 μm, so that the mold can be fabricated with low-cost xurography. Scanning Electron Microscope images of (**ii**) planar and (**iii**) cross-sectional views of the device show that the mold produces smooth channel walls. (**b**) The largest beads (green) are focused at earlier turns of the device, migrate closest to the inner channel wall and are collected at the first outlet. Focused particle streams are spaced closely together.

**Figure 2 micromachines-11-00514-f002:**
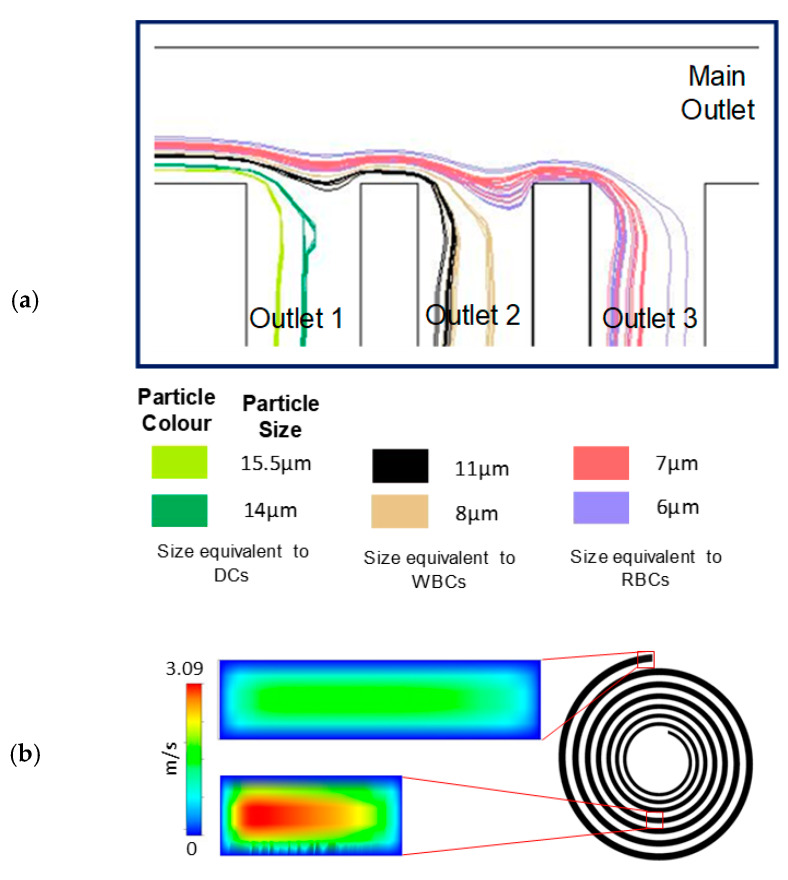
Computational fluid dynamics simulations were used in the design of our device. (**a**) Simulation of particle streamlines verify that particles larger than 13 μm enter collection Outlet 1, particles between 8 and 12 μm enter collection Outlet 2 and particles between 6 and 7 μm enter Outlet 3. (**b**) Simulations verify that Dean forces are more rapidly reduced in the spiral with expanding channel widths.

**Figure 3 micromachines-11-00514-f003:**
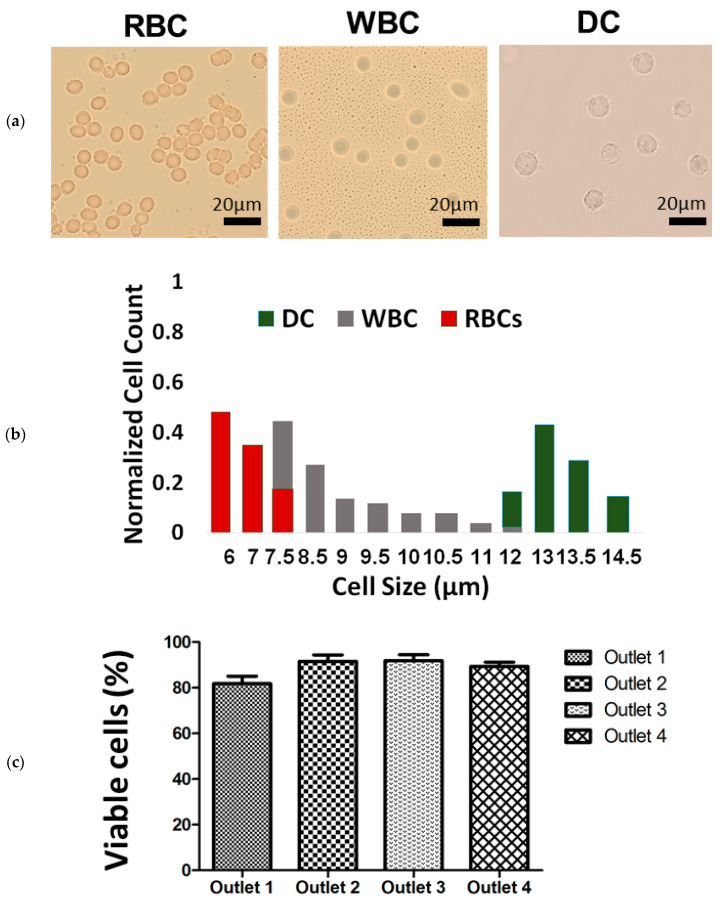
Our device was designed to perform a blood-cell separation assay. (**a**) Images of red blood cells, white blood cells and dendritic cells, together with (**b**) the distribution of cell sizes, suggest that blood-cell types can be separated reasonably according to size. (**c**) Inertial focusing and high flow rates do not harm cells, because cell viability in all four collection outlets exceed 80%.

**Figure 4 micromachines-11-00514-f004:**
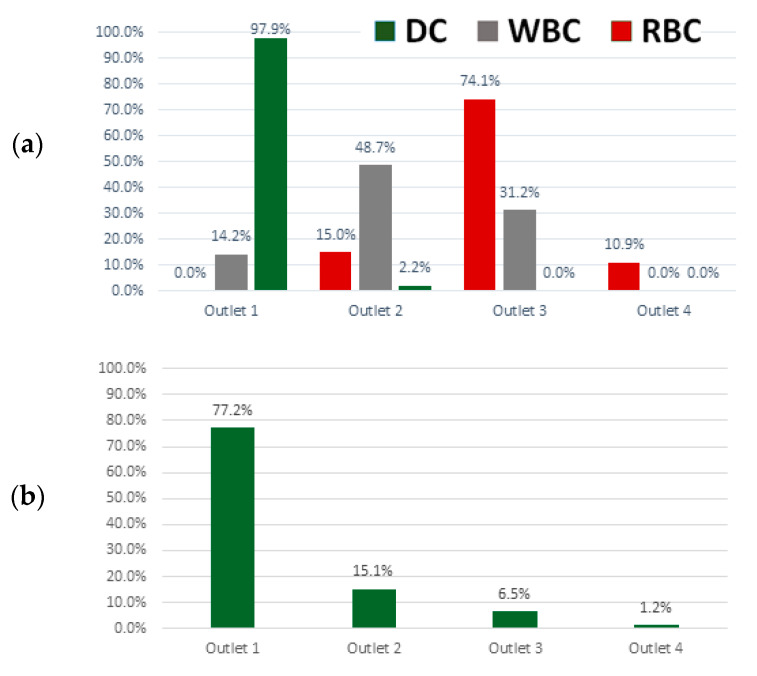
Evaluation of blood-cell-separation efficiency at a flow rate of 1.9 mL/min: (**a**) the majority of dendritic cells are collected in Outlet 1, white blood cells in Outlet 2 and red blood cells in Outlet 3 when the spiral microfluidic device is tested with fluid containing a single cell type only at 100× dilution. (**b**) The majority of dendritic cells are still collected in Outlet 1, although with lower efficiency, when tested with fluid containing all three types of blood cells.

**Figure 5 micromachines-11-00514-f005:**
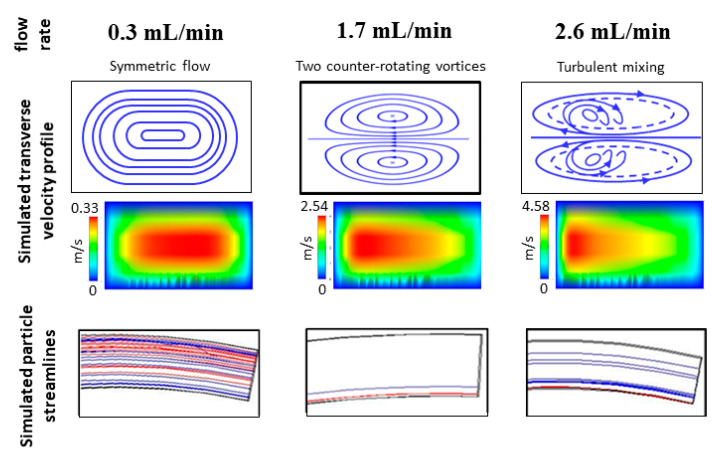
Simulation studies at three different fluid velocities show the spectrum of particle behaviors in the spiral microfluidic device. At 0.3 m/s, there are insufficient dean and inertial lift forces, so particles are not focused nor separated. At 1.7 m/s, the lateral Dean forces are slightly less than inertial lift forces, and this supports particle focusing and fractionation. At 2.6 m/s, Dean forces dominate, and particle streams are defocused due to lateral mixing.

**Figure 6 micromachines-11-00514-f006:**
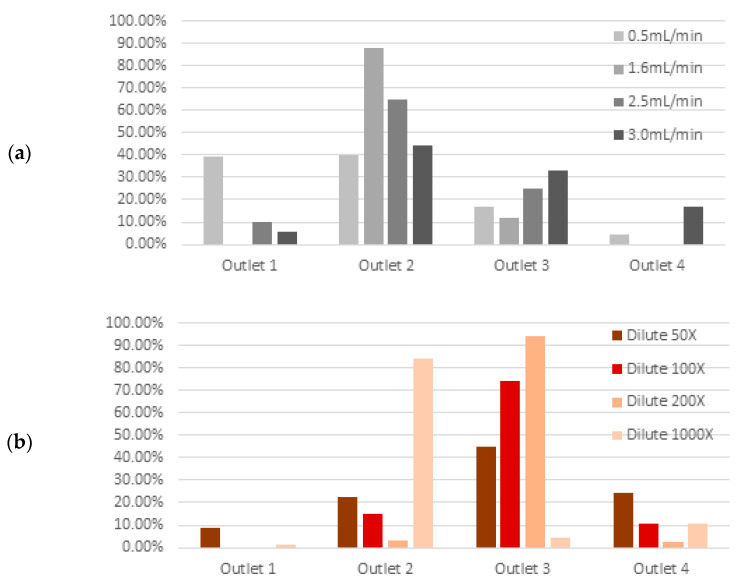
Blood cell separation efficiency, (**a**) using only white blood cells at 100× dilution while varying flow rate, an (**b**) using only red blood cells at 1.9 mL/min while varying concentration.

**Table 1 micromachines-11-00514-t001:** Channel widths of molded PDMS devices can be fabricated with tolerance of less than 5%.

Position	Channel Depth (µm)	Desired Channel Width (µm)	Fabricated Channel Width (µm)	Error (%)
Turn 2	100	200	208	4
Turn 2	100	200	193	3.5
Turn 4	100	400	390	2.5
Turn 4	100	400	412	3
Turn 6	100	600	589	1.8
Turn 6	100	600	608	1.3

**Table 2 micromachines-11-00514-t002:** Key dimensionless fluidic parameters at selected loci on the spiral, at a constant flow rate of 1.8 mL/min. With an expanding channel width, particle Reynolds number and fluid velocity will vary along the spiral channel.

Spiral Position	Fluid Velocity, Uf (m/s)	Reynolds Number, Re	Dean Number, De	Particle Reynolds Number, Rp		
				7 μm	10 μm	15 μm
Inlet	1.50	200	36.51	0.60	1.13	2.53
Turn 1	1.0	150	27.09	0.36	0.67	1.5
Turn 2	0.86	133	22.84	0.30	0.55	1.24
Turn 3	0.75	120	19.75	0.25	0.47	1.05
Turn 4	0.67	109	16.91	0.22	0.41	0.92
Turn 5	0.60	100	14.61	0.19	0.36	0.81
Turn 6	0.55	92	12.72	0.17	0.32	0.73
Turn 7	0.50	86	11.16	0.16	0.29	0.66

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
