# Peer review of "Label-Free, High-Throughput Assay of Human Dendritic Cells from Whole-Blood Samples with Microfluidic Inertial Separation Suitable for Resource-Limited Manufacturing"

_micromachines, 2020, doi:10.3390/mi11050514_

Round 1

Reviewer 1 Report

The authors develop a inertial separation spiral microchannel device to fractionate blood cells by size. The channels can have more than 200 micron width, which is a dimension suitable to be manufactured by xurography. The device is used to isolate dendritic cells from whole-blood samples. I recommend publications after the authors address the following issues:

  1. Particle fractionation in the device should depend on non-dimensional numbers, so the authors need to calculate and present the non-dimensional numbers relevant to the case studied. At least the following non-dimensional numbers should be given: particle-size/channel_width, Reynolds number, particle Reynolds number, Dean number. The average flow velocity should also be calculated.

  1. The authors should discuss what would happen if they operated at a different flow rate. This change would change the non-dimensional numbers above. Please discuss how scale-up of scale-down could influence separation and if there are sufficient degrees of freedom to compensate scale-up (or scale-down).

  1. Photos of the cells should be included in the paper so that the reader can see the differences in size.

  1. The authors should comment on the fact that cells are flexible and particles are rigid. This difference may imply different behavior.

  1. Detailed microscope photos of the channel should also be given so that the reader can see the fabrication details.

  1. Table fonts are too small.

Author Response

We thank the anonymous reviewer for the insightful comments which has helped us improve the quality and clarity of the manuscript

  1. Particle fractionation in the device should depend on non-dimensional numbers, so the authors need to calculate and present the non-dimensional numbers relevant to the case studied. At least the following non-dimensional numbers should be given: particle-size/channel width, Reynolds number, particle Reynolds number, Dean number. The average flow velocity should also be calculated.

The computed, non-dimensional numbers and the fluid velocity at various positions in the spiral microfluidic device (same angular position as the inlet but at different turns) are reported in the new Table 2.

  1. The authors should discuss what would happen if they operated at a different flow rate. This change would change the non-dimensional numbers above. Please discuss how scale-up or scale-down could influence separation and if there are sufficient degrees of freedom to compensate scale-up (or scale-down).

Simulation results are provided for transverse velocity profile (indicative of Dean force) and particle streamlines in Figure 5 at different flow rates. We also report in the results that simulations show that a suitable range of flow rates are 1.5mL/min to 1.9mL/min. Results from experiments with blood cells at different flow rates are presented in the new Figure 6a and verify these findings. Discussion section has been expanded and paragraph 7 discusses the scaling behavior.

  1. Photos of the cells should be included in the paper so that the reader can see the differences in size.

We have included brightfield microscope images of red blood, white blood and dendritic cells and an experimentally-measured distribution of blood cell sizes in the new Figure 3.

  1. The authors should comment on the fact that cells are flexible and particles are rigid. This difference may imply different behaviour.

We have inserted brief referenced comments in paragraph 8 of the discussion section.

  1. Detailed microscope photos of the channel should also be given so that the reader can see the fabrication details.

Scanning electron microscope images of the fabricated spiral device channels have been included in Figure 1. Table 1 also summarizes measurements of fabricated channel width at selected locations on the spiral device in comparison to the designed geometry.

  1. Table fonts are too small.

For clarity and due to the increased number of figures, tables have been removed but a summary of the table data are still presented in graphical form in Figure 3c and Figure 4b

Reviewer 2 Report

The manuscript by Yousuff et al describes the development of a spiral-based microfluidic device for the isolation of dendritic cells using inertia as driving force. The microfluidic device is fabricated using xurography, a low-cost and simple methodology based on CNC micromachining. The microfluidic device is first validated using fluorescent beads with different sizes and colors as a testbed, and then, employed for the separation of human dendritic cells. Even though the use of microfluidic-based inertial separation is well and broadly reported, I find this paper interesting, mainly because of the used fabrication technology. However, and unfortunately, the provided results are scarce, which makes this paper not suitable for publication as it is. The authors should provide much more solid results to make this paper acceptable for publication. Other minor comments are the followings:

  • General comments: In some sections the text is difficult to read (English grammar, long sentences...). Please, check and correct it.
  • Authors should stress about the current methodologies to isolate dendritic cells (or other cell types) in the Introduction, and the need/advantage of using microfluidics.
  • The dimension, shape,… of the microfluidic spiral should be provided in the Materials and Methods section and not in the Results.
  • Authors should test their device for the isolation of other cell types.
  • Authors should provide the simulation results as a new figure panel in the manuscript.
  • Please, add the scale bars in all the figures.
  • Figure 1B, bottom left fluorescence panel: Why the particle streams are located in the upper region of the channels? If inertial forces take place, they should be located near the bottom. Please, clarify.
  • Authors should provide more experimental evidences about the performance of their device. For instance, how the efficiency of isolation would change upon the modification of: number of turns, flow rate, cell concentration,…etc.

Author Response

We thank the anonymous reviewer for the insightful comments which has helped us improve the quality and clarity of the manuscript

  1. In some sections the text is difficult to read (English grammar, long sentences...). Please, check and correct it.

Long sentences in the introduction and discussion have been rephrased and shortened to improve clarity

  1. Authors should stress about the current methodologies to isolate dendritic cells (or other cell types) in the Introduction, and the need/advantage of using microfluidics.

Paragraphs 4 and 5 in the Introduction have been added to describe current methodologies to isolate and enrich dendritic cells and the advantages of a microfluidics POC assay.

  1. The dimension, shape,… of the microfluidic spiral should be provided in the Materials and Methods section and not in the Results.

The dimensions and designed geometries of the microfluidic spiral device are now described in Section 2.1 in the Materials & Methods section together with new information on the simulation software and algorithms.

  1. Authors should test their device for the isolation of other cell types.

We have introduced a new Figure 4 which reports the separation efficiency achieved for red blood cells and white blood cells.

  1. Authors should provide the simulation results as a new figure panel in the manuscript.

Computational fluid dynamic simulations of particle streamlines at the collection outlets and the transverse velocity profile are included in the new Figure 2. Streamline simulations at the outlet show the range of particle sizes captured in each outlet while velocity profile simulations show that transverse Dean forces are only significant midway in the spiral device but degenerate near the outlet. We have also included simulations of particle streamlines and transverse velocity profile simulations with varying fluid velocities in new Figure 5. These simulations illustrate representative scenarios at different fluid velocities when particles are not focused (0.3m/s), when particle are focused and fractionated (1.7m/s) and when particles are defocused by transverse mixing (2.6m/s).

  1. Please, add the scale bars in all the figures.

Scale bars have been added to all images

  1. Figure 1B, bottom left fluorescence panel: Why the particle streams are located in the upper region of the channels? If inertial forces take place, they should be located near the bottom. Please, clarify.

We apologize for the confusion. Focused particles should congregate on the side of the inner wall (wall with smaller radius). Figure 1b has been edited to improve clarity. The crop box showing particle streams in all 7 turns have been re-positioned vertically. Arrows now highlight turn 1 near the inlet where particles are unfocused and channel width and radius are smallest and turn 7 near the outlet where particles are focused and channel width and radius are largest.

  1. Authors should provide more experimental evidences about the performance of their device. For instance, how the efficiency of isolation would change upon the modification of: number of turns, flow rate, cell concentration,…etc.

The isolation efficiency of blood cells as a function of flow rate and cell concentration are report in the new Figure 6. Figure 1b implicitly shows that between 6 to 7 turns is the minimum length to achieve good isolation efficiency for all particles sizes in the design range. Adding additional turns is not expected to change the results materially until fluid Reynolds number drops beyond a threshold to sustain inertial drag forces to maintain particle focusing.

Round 2

Reviewer 1 Report

The paper can now be published

Reviewer 2 Report

The authors addressed satisfactorily my main comments related to the revised manuscript and therefore, I recommend its publication.